# Different Distribution Patterns of Hoverflies (Diptera: Syrphidae) and Bees (Hymenoptera: Anthophila) Along Altitudinal Gradients in Dolomiti Bellunesi National Park (Italy)

**DOI:** 10.3390/insects13030293

**Published:** 2022-03-15

**Authors:** Daniele Sommaggio, Livia Zanotelli, Enrico Vettorazzo, Giovanni Burgio, Paolo Fontana

**Affiliations:** 1Department of Agricultural and Food Sciences—DISTAL, University of Bologna, 40127 Bologna, Italy; giovanni.burgio@unibo.it; 2Fondazione Edmund Mach—Centro Trasferimento Tecnologico, 38057 Pergine Valsugana, Italy; livia.zanotelli@fmach.it (L.Z.); paolo_api.fontana@fmach.it (P.F.); 3Dolomiti Bellunesi National Park, 32032 Feltre, Italy; e.vettorazzo@dolomitipark.it

**Keywords:** Syrphidae, Anthophila, hoverflies, bees, pan trap, altitudinal distribution, Italian national parks

## Abstract

**Simple Summary:**

Hoverflies and bees play a key role in plant pollination. The increasing concern about pollinator reduction forces the planning of a sampling monitoring scheme to evaluate the change in the populations of these important insects. The present research provides baseline data about the distribution of hoverflies and bees in the Dolomiti Bellunesi National Park (Northeastern Italy). The hoverfly community shows a unimodal distribution with peak at middle elevation, while bees display a linear reduction in richness and abundance with increasing altitude. Both hoverfly and bee β-diversity at high altitude is dominated by species turnover more than by nestedness.

**Abstract:**

Hoverflies (Diptera: Syrphidae) and bees (Hymenoptera: Anthophila) are two key taxa for plant pollination. In the present research, the altitudinal distribution of these taxa was studied along two gradients (elevation range: 780–2130 m) in the Dolomiti Bellunesi National Park (Northeastern Italy). Pan traps were used as a sampling device to collect both hoverflies and bees. Other than altitude, the effect of landscape complexity and plant diversity were considered as potential predictors of hoverfly and bee richness and abundance along the two gradients. A total of 68 species of hoverflies and 67 of bees were collected during one sampling year, confirming the efficacy of pan traps as a sampling device to study these taxa. Altitude was the main variable affecting both hoverfly and bee distribution. The two taxa show different distribution patterns: hoverflies have a unimodal distribution (richness and abundance) with peak at middle altitude (1500 m), while bees have a monotonic decline (richness and abundance) with increasing altitude. Both hoverfly and bee populations change with the increasing altitude, but the change in hoverflies is more pronounced than in bees. Species turnover dominates the β-diversity both for hoverflies and bees; therefore, the hoverfly and bee communities at higher altitudes are not subsamples of species at lower altitude but are characterized by different species. This poses important conservation consequences. Some rare species, typical of an alpine habitat were recorded; the present research represents important baseline data to plan a monitoring scheme aimed at evaluating the effect of climate change on pollinators in these fragile habitats.

## 1. Introduction

Since the mid-1990s the scientific community has become aware of the increasing decline in pollinator biodiversity, commonly referred to as the “pollination crisis” [1,2,3]. Biodiversity loss of bee species, which represent one of the most investigated taxa, have been well-documented in several countries [4,5,6,7,8,9]. Even if less investigated, a similar trend has also been recorded for other pollinator taxa, such as Rhopalocera (Lepidoptera) [10] and Syrphidae (Diptera) [11,12]. The pollination crisis has deeply concerned the whole scientific community not only regarding conservation issues but also because of the great role that the ecosystem service of pollination plays in the world economy: IPBES [13] estimates that 5–8% of world crop production in 2015 is pollinated by animals, which corresponds to an annual value of USD 235–577 billion.

There are several causes that can concur to the “pollination crisis”. An increasing amount of data demonstrates the detrimental impact of pesticides on pollinators [14,15,16,17], which led to restrictions of neonicotinoids and fipronil in the European Union [18,19]. Loss of habitat and fragmentation are additional causes of bee decline due to a reduction in floral and nesting resources, together with a reduction in tolerance thresholds against disease and pesticides (e.g., [15,20,21]). Climate change is considered another major cause of pollinator decline [20,22]. The effect of climate change may be particularly strong in plant–pollinator interactions, where some degree of synchronization is necessary to ensure pollination. Available data on this topic are sometimes conflicting, and to date, it is not fully understood whether climate change can generate negative consequences due to this desynchronization, leading to a reduction in pollination [23,24,25].

Alpine habitats are considered as particularly exposed to climate change [26,27,28]. According to Engler et al. [29], 36–55% of alpine species and 31–51% of subalpine species could lose more than 80% of suitable habitat by 2070 to 2100. The shift towards higher altitudes is one of the main responses of plants and animals to increasing temperatures, but this adaptation may be impossible for alpine species living at the peak of mountains, with the consequence of a greater fragmentation of populations, leading to an increasing risk of extinction [30,31,32].

Moving towards higher altitudes results in changes in the pollinator community. According to Bergmann’s rule, insects should be bigger at higher altitudes because a larger size reduces the ratio between body surface and volume, allowing for a more effective control of heat loss [33]. However, the distribution of taxa along altitudinal gradient did not always confirm this rule [34,35,36,37]. At higher altitudes, the presence of Hymenoptera usually decreases, with a parallel increase in Diptera [38,39,40,41,42]. In particular, moving towards higher elevation, two main consequences, which are often concomitant, can be evinced: a change in physical conditions, including temperature, oxygen partial pressure, and insolation, and changes at landscape levels, such as a reduction in wood coverage [43,44].

In order to better understand how different focal taxa will respond to future climate change in Alpine habitats, it is of fundamental importance to have biodiversity data banks and recordings in order to distinguish the effects of physical conditions from that of landscape shifts. In addition to evaluating the trends of pollinators in the medium and long term, it is necessary to plan monitoring schemes repeated over time. This knowledge is particularly crucial to plan conservation interventions and to understand the effects of climate change on biodiversity, including pollinator species. The availability of data belonging to different geographical areas can allow comparative analyses, thus achieving an understanding of these complex phenomena.

In 2005, the Gran Paradiso National Park developed a biodiversity monitoring project in the Alpine environment that also involved some regional parks of the Western Alps. Since 2013, the initiative has been extended to the other National Parks of the Italian Alps (Val Grande, Stelvio and Dolomiti Bellunesi). The project, called the Biodiversity Project and still active, included a standardized monitoring of different taxa (Aves, Orthoptera, Coleoptera Carabidae and Staphylinidae, Lepidoptera Rhopalocera, Hymenoptera Formicidae and Araneae), along altitudinal transects. In 2020, the Dolomiti Bellunesi National Park also extended the research to Anthophila (Hymenoptera) and Syrphidae (Diptera). The role of bees as pollinators has been largely recognized because they are one of the main taxa in plant pollination [45,46,47]. Although hoverflies receive less attention than bees as pollinator, recently, increasing data have emphasized the importance of these flies in pollination, in some cases being more important than bees (e.g., at higher altitudes and mountain habitats) [48,49,50,51].

The objectives of the present study are the following:To evaluate the altitudinal distribution of two important pollinator taxa, i.e., Hymenoptera Anthophila (bees) and Diptera Syrphidae (hoverflies), also including the phenological patterns;To compare the effects of landscape and altitude as main factors affecting species distribution.

The present paper is focused on the results of bee and hoverfly sampling, including the influence of environmental and climate factors on diversity of these taxa, paying particular attention to the trends of abundance and species richness monitored at different altitudes.

## 2. Materials and Methods

### 2.1. Study Site

The present research was conducted in 2020 along two altitudinal gradients in the oriental part of Dolomiti Bellunesi National Park in the Veneto Region (Figure 1). The two gradients have been previously selected for the Biodiversity Project monitoring. The altitudinal gradients are as follows:Vette Feltrine (VF): This gradient starts at 780 until 2130 m a.s.l.; 8 circular plots (100 m radius) were selected at around 200 m altitude difference (Table 1).Mount Grave (MG): This gradient is shorter than previous one, starting at 1080 m and ending at 1375 m; 3 circular plots were selected at altitude differences of around 200 and 100 m (Table 1).

For each transect, the following predicting variables were measured: altitude (m), temperature (°C), slope (°), landscape complexity, and vegetation diversity. Altitude was measured by GPS and confirmed with the Regional Technical Map. Temperature was recorded using a remote sensor of temperature (Thermochrom i-Button), placed at the centre of each plot. The sensor was placed at a high of 1 m inside a plastic white tube (20 cm long and 10 cm diameter) to protect it from direct solar radiation. The tube had holes to ensure the free circulation of air around the sensor. The temperature was recorded hourly throughout the day. The mean minimum temperature was calculated from the beginning of June to the end of September for each sampling point. During the sampling year (2020), some remote sensors had malfunctions, and for this reason, we have decided to use the average values of the temperatures detected by the sensors in the previous two years. Clinometer (Suunto PM5–1520 PC) was used to measures slope at the centre of each plot. Landscape complexity was evaluated using GIS tools; the map used was the CTR–Regional Technical Map, Veneto Region (https://www.regione.veneto.it/web/ambiente-e-territorio/carta-tecnica-regionale, accessed on 31 January 2022). Each plot consisted of a circular area with a radius of 100 m. The type and extension of vegetation was evaluated in the circular area using ArcMap 9.3 Esri. The percentage of the following habitats were calculated: percentage of woods, percentage of shrubs, percentage of grassland, percentage of scree. The percentage of peat bog was very small (only 7% in only one plot) and not included in the following analysis. The plant species and their coverage were evaluated in 2013 in three points (inside each plot) following the minimum area approach [52]. For the present research, the number of species detected inside the plot (total plant species) and the number of plant species pollinated by insects has been used as plant diversity estimator. Even if the plant diversity was monitored in 2013, the plots along the two gradients were visited each year within the Biodiversity Project, and no consistent change in vegetation was observed between 2013 and 2020.

### 2.2. Sampling Design

Both hoverflies and bees were sampled by coloured pantraps: white, blue, and yellow, as suggested in current literature [53,54,55]. Pan traps and active netting along transects are considered the most suitable methods to sample bees for their efficiency, repeatability, and standardization [53,56,57,58]. We decided to use pan traps instead of entomological nets in order to avoid a potential sampling bias due to the collector; furthermore, given the high climate variability over time along the gradient, we preferred a sampling method that would reduce the field time. Finally, the steep slope in some sampling points along the gradient would have made sampling by net very laborious and time consuming.

In each plot, 5 sampling points were selected; in each sampling point, a stake was placed, each one supporting 3 white bowls (diameter: 30 cm; height: 5 cm), at about 60 cm from the ground, at similar level than most flowers (Figure 2). The inside of bowls was painted blue, white, or yellow UV-reflecting paints (Sparvar Leuchtfarbe, Spray-Color GmbH, Merzenich; item numbers 3107, 3108, 3104, respectively) [53]. The five sampling points were set at a distance of 25 m along the diameter of the plot. The distance between sampling points was chosen in order to ensure the independence of each group of bowls [59].

Pan traps were activated each month from May to September 2020 for a total of five sampling dates; however, due to adverse climate conditions at high altitude, the plots in VF gradient from 1200 to 2100 m were sampled only from June to August. Instead, MG gradient was regularly sampled from May to August. Pan traps contained water and a few drops of surfactant. In each sampling, the traps remained in the field for 2 days in good weather conditions, avoiding sampling on rainy days. The insects collected from the five plots within each transect were pooled together. Bees and hoverflies were identified at species level using the following key:-Syrphidae: [60,61,62];-Anthophila: [63] for genera *Bombus* and *Psithyrus*; [64] and references included for other genera.

For the identification of five specimens of *Bombus* (Apidae), for which morphological characters were not informative, molecular identification was carried out, using cytochrome oxidase gene (COI) [65,66]. The details are reported in Appendix A.

### 2.3. Statistical Analysis

The collinearity between predicting variables was evaluated by Spearman linear correlation (*p* < 0.05). In accordance with this preliminary test, the following pairs of variables were correlated (Appendix A):Temperature was correlated with altitude;Percentage of grassland was inversely correlated with percentage of woods;Percentage of scree was correlated with altitude;Percentage of scrubs was correlated with percentage of grassland;Slope was correlated with plant species pollinated by insects;Total plant species was correlated with plant species pollinated by insects.

Temperature and percentages of scree, scrubs, and plant species with entomophilic pollination were excluded. Only the following predicting variables were selected: altitude, slope, type of gradient (VF or MG), total plant species, percentage of wood, or percentage of grassland. The last two variables were alternatively used in multivariate analysis, selecting those with a better predictive power.

In order to detect potential effects of the selected predicting variables on hoverfly and bee community, an ordination analysis was used. A distance-based Redundancy Analysis (db-RDA) was applied to hoverfly and bee matrices of abundance after Hellinger transformation [67,68]. To test the significance of canonical axes, an ANOVA permutation test was applied using the function anova.rda in the vegan package in R. The function was applied to both the full model and, using the setting by = “terms”, to each independent variable [69].

A db-RDA was also applied to ecological categories of hoverflies and bees. For hoverflies, the following categories were used: larval trophic habitus (three categories: zoophagous, saprophagous, phytophagous); larval microhabitat association (five categories: tree foliage, root, herb layer, trunk trees); dimension (three categories: small (6–9 cm), medium (9.5–12 cm), large (>12 cm)). The information about the ecological category of each species was obtained by [12]. For bees, the following categories were selected: diet breadth (three categories: oligolecty, polilecty, social parasite), nest position (three categories: soil, vegetation, holes), dimension (three categories: small (5–10 cm), medium (10.5–15.5 cm), large (>15.5 cm)). For each transect, the total number of species belonging to these categories was pooled.

To investigate the influence of elevation on bee and hoverfly richness and abundance, linear and polynomial quadratic correlations were performed, using elevation as a predictor variable. We applied both models to the three sampling months (June, July, and August) and on the whole period (total). We selected the model with the higher R^2^. A Durbin–Watson test was performed to evaluate the independence of residuals.

To evaluate the relationship between bee and hoverfly community changes against elevation, two approaches were used. A Mantel test was applied to evaluate whether the diversity in hoverfly and bee community changes according to increasing altitudinal distance [70]. We analysed the association between distance matrices of transect altitudes and Jaccard dissimilarity for hoverfly and bee communities with 10,000 permutations. The test of ANCOVA (Analysis of Covariance) was applied to the two correlations (hoverfly dissimilarities vs. altitudinal distances and bee dissimilarities vs. altitudinal distances) in order to evaluate the presence of differences in slopes and/or intercept. Shapiro and Levene tests were previously performed to evaluate normality and homogeneity of variance, respectively. Finally, the regression lines were compared with the aim to understand the interaction between the two taxa in relation to the altitude variable [71].

Beta-diversity partitioning: The object of this analysis was to separate species turnover (or species replacement) and nestedness (or difference in richness) [72,73]. Therefore, this analysis was used to understand if bee and hoverfly communities at high altitudes can be considered as a subpopulation of low-altitude populations (nestedness dominance) or a population dominated by different species (species turnover dominance). Βeta-diversity partitioning was applied on incidence (presence/absence) matrices of bees and hoverflies [72,74]. The turnover of species (βsim) was calculated using Simpson dissimilarity and nestedness (βnes) using Sørensen dissimilarity index; the sum of these two components is equal to total beta diversity (βsør): βsør = βnes + βsim.

All statistical analyses were employed in R (version R 4.1.0), using the following packages: Hmisc (correlation matrix) [75]; vegan (db-RDA, Mantel test) [69]; betapart (β-diversity partitioning) [76]; all graphs were developed using ggplot2 and ggpubr [77].

## 3. Results

The pan traps collected 768 hoverfly and 1237 bee specimens, belonging to 68 and 67 species, respectively (Appendix A). Figure 3 and Figure 4 list the species recorded, including absolute abundance at different altitudes. The most common hoverfly species were *Eristalis tenax* (20.4%), *Xylota segnis* (13.9%), and *Merodon rufus* (10.7%). The more abundant bee species were *Apis mellifera* (35.1%), *Lasioglossum calceatum* (10.6%), and *Bombus terrestris* (9.2%). A total of 29 (3.9% of total) hoverfly and 68 (5.5%) bee specimens were collected in May and September only at low-altitude sites, and for this reason, they were excluded from the following analysis.

db-RDA was performed separately using hoverfly (68 species) and bee (67 species) matrices of abundances, including five environmental variables (Figure 5). For both hoverflies and bees, the best model was the one including the altitude, the percentage of wood, the slope, the type of gradient, and the total plant species. This model explained 66.7% of the total variability for hoverflies and 66.5% for bees. The hoverfly family was significantly influenced by both the type of gradient (F = 3.53, *p* < 0.001) and the altitude (F = 3.12, *p* = 0.011). No effect of the percentage of wood, slope, and total plant species was detected on hoverfly population. Similar results were found for bees: A significant effect of the type of gradient (F = 3.57, *p* < 0.001) and altitude (F = 2.3, *p* = 0.006) was recorded; in addition, the slope also showed a significant effect (F = 1.93, *p* = 0.017), while no influence of the percentage of wood and the total plant species was observed.

No significant model was obtained when db-RDA was applied to the ecological categories of hoverflies and bees. The altitude, type of gradient, slope, percentage of wood, and total number of plants had no effect on hoverfly and bee categories; in particular, no trend of increasing size was correlated with altitude, either for hoverflies or bees.

Since the different types of gradient (Mount Grave and Vette Feltrine) had a significant effect on both hoverfly and bee population, in the subsequent analysis, only data from Vette Feltrine were considered, removing Mount Grave. Indeed, the Vette Feltrine plots showed a higher association with the “altitude” vector (Figure 5).

Since “altitude” was the predicting variable which mostly explained data pattern, a specific analysis was performed for this factor. Both total abundance and species richness of hoverflies and bees were significantly correlated with the altitude (Table 2). However, while the correlation was linear (negative) in bees, hoverfly showed a curvilinear polynomial correlation of quadratic type (Figure 6). The same trends were also obtained considering the data collected in each sampling month, even if a significant correlation was detected only in one month both in hoverflies (July both for species richness and abundance) and bees (August only for species richness) (Table 2, Figure 7). Therefore, the highest abundance and richness in the two taxa were recorded at different altitudes, namely, at medium altitudes for hoverflies (around 1500 m) and at lower ones (around 800 m) for bees.

For both hoverflies and bees, the Mantel test revealed a significant correlation between community dissimilarity against the Euclidean distance of altitude among sites (hoverflies: Mantel r = 0.48, *p* = 0.004; bees: Mantel r = 0.47, *p* = 0.016). This means that as the altitudinal distance between the sites increases, the dissimilarity between the communities of the taxa also enhances (Figure 8).

According to Shapiro and Levene tests, data assumptions are satisfied for the application of ANCOVA. The model performed with altitude and taxa (hoverflies and bees) was significant for both variables (F_(1,53)_ = 11,254, *p* = 0.001 and F_(1,53)_ = 21,846, *p* < 0.0001, respectively), but the interaction between altitude and taxa was not significant. Parallelism test showed no difference between correlation slopes (Figure 8) while elevations of the lines were significantly different (*p* < 0.0001); in other words, the change in hoverfly population along increasing altitude is more pronounced than that recorded for bees.

β-diversity partitioning showed that differences between transects were mainly due to species turnover, which accounted for 89% of total variation (compared to 11% for nestedness) for hoverflies and 91.2% of turnover (8.8% of nestedness) for bees.

## 4. Discussion

The number of species recorded was 68 for hoverflies and 67 for bees, making up 13% and 5.6% of total Italian fauna, respectively. *Eristalis tenax* was the dominant hoverfly species, a taxon that belongs to a genus considered among the most efficient pollinators for its peculiar biological and morphological traits (high mobility, low ecological requirements, dense hairiness) [50]. Some species are of conservation interest because they are considered rare. Among them, *Brachypalpus chrysites* (Syrphidae) is a saproxylic species, and this is the second recorded in North Eastern Italy (Sommaggio, unpublished data); *Trychopsomyia joratensis* has predator larvae, which probably develop in humid forests, their record is the second one in Italy [78]. *Cheilosia laeviseta* is a species associated with mountain grasslands, found only in the Pyrenees, Alps, and Balkans [12]: in the present study, it was recorded only at high altitudes. Three bee species, *Andrena hattorfiana*, *A. ovatula*, and *Lasioglossum laticeps*, are considered as Near Threatened in [9], while *Bombus inexpectatus* is included in bee red list as Endangered. *Bombus inexpectatus* is associated with high mountain habitats of the Alps and Pyrenees; its highly fragmented distribution exposes this species to the risk of local extinction.

Several investigations compared the efficiency of different methods in sampling hoverflies and especially bees [53,56,57,58]. Pan traps and entomological nets along transects proved to be the most efficient methods, though not fully exhaustive. Malaise traps have been suggested as the standard sampling device to study hoverfly population [79], although pan traps are also considered a proper sampling method where a large number of replicas are carried out [80]. However, some criticisms have been expressed towards the use of pan traps [81]. Pan trap and active netting collections reported different assemblages of species; in addition, pan traps seem to be more efficient in the absence of floral resources, in opposition to active netting [82]. Also in the present study, some bias in hoverfly and bee sampling due to the use of pan traps may be likely. For this reason, the list of species resulting from this study may not be complete. The integration of pan traps with active netting could be a solution for a more exhaustive sampling of bee and hoverfly populations, but active netting along a transect of over 1300 m in altitude is almost impossible for a single sampler to perform, ensuring the same climatic conditions. Active netting would have determined additional bias by the sampler, sampling day, and time of day, seriously affecting the sampling.

The effect of altitude on species distribution has been studied for a long time. Two patterns of animal distribution along an altitudinal gradient have been more frequently detected: (1) a monotonic decline in species richness with increasing altitude and (2) a unimodal peak at middle elevations [83,84,85,86]. Little research has focused on Diptera and Hymenoptera, however, both distribution patterns were recorded in families belonging to these taxa: A unimodal peak at middle elevation was recorded in Anthophila [87], Tabanidae [88], Simuliidae [89], and Empididae [90]; instead, a monotonic decline was observed in Anthophila [91], Vespidae [92], and Tephritidae [93].

The altitudinal range, geographic area, and sampling device are variables affecting the distribution of taxa on an altitudinal gradient [94,95]. In the Swiss Alps, Beck et al. [96] found diverse patterns in moths in different seasons: a monotonic decline in spring and autumn and a unimodal peak at middle elevation in summer. At specific conditions, additional patterns have been found; for example, at low altitude gradients (400–700 m), Baumann et al. [97] found richness and abundance of wild bees and abundance (but not richness) of hoverflies increasing with altitude. The same pattern (increasing diversity following altitude increase) has been detected also in Formicidae in arid conditions [98].

Peters et al. [86] analysed the altitudinal distribution of 25 plant and animal taxa along a 3.7 km gradient in Kilimangiaro Mountain: A monotonic decrease with altitude was detected in six invertebrate taxa (including wild bees), while a unimodal peak at middle elevation was observed in six taxa (including hoverflies). Data collected in Dolomiti Bellunesi National Park confirm that different taxa may exhibit different distribution patterns along an altitudinal gradient; in particular, hoverflies have a unimodal distribution with a peak around 1400–1500 m (at least in the studied period), while wild bees display a monotonic decrease with increasing altitude.

Despite the large amount of data available on organism distribution along altitudinal gradients, the drivers that determine these patterns are still far from being understood [99,100,101]. Several hypotheses have been advanced; traditionally, abiotic drivers (mainly temperature and water availability) have been largely evaluated as main factors affecting the biodiversity on mountains [86,101]. The “productivity hypothesis” focuses mainly on the different distribution of resources with altitude [102,103,104]. Plant richness and differences in habitats were the biotic drivers widely suggested to explain biodiversity distribution [105]. The “area hypothesis” stated that spatial constraints reduce biodiversity with increasing altitude [106]. Finally, the Mid Domain Effects suggests that stochastic geometric phenomena can be responsible for different distribution of species; this theory has been largely used to explain the unimodal distribution [85,107,108,109].

In the present study, abiotic factors (in particular, temperature, strongly correlated with altitude) seem to be more important in affecting hoverfly and bee populations along the altitudinal gradient. The biotic parameters (in particular, the plant diversity and landscape composition) seem to have no effect, or a marginal one, on hoverfly and bee populations in the altitudinal gradient examined in the present study. Viterbi et al. [110] found that invertebrate (spiders, ground and rove beetles, and butterflies) distribution along an altitudinal gradient was affected by temperature, while bird community was influenced mainly by habitat structure. Peters et al. [86] found different potential drivers in various taxa, but temperature proved to be the main one at total biodiversity level. The different pattern observed in hoverflies and bees in our study may be due to a stronger effect of temperature on bees than in hoverflies or a Mid Domain Effect on hoverflies. Bee populations at high altitudes were dominated by *Bombus* species (50% of all species at high altitude belonged to this genus); instead, more hoverfly genera were recorded at high altitude (e.g., *Cheilosia*, *Merodon*, *Eristalis*, *Volucella*). Further studies are necessary to better understand and explain the differences observed between these two taxa.

Despite the fact that hoverflies and wild bees showed different altitudinal patterns, in both cases the change in population structure was dominated by diversity turnover more than nestedness [72,110,111,112]. This mean that for both taxa, population at high altitude is not a subset of that of lower altitude, but different species are present at different altitudes. In the present study, *Cheilosia derasa*, *C. grisella*, *C. laeviseta*, *Merodon aeneus*, *Platycheirus manicatus*, and *Sphaerophoria fatarum* are species found only at high altitudes; in the Alps, these species are typical of montane grasslands [12]. The dominance of species turnover moving with altitude has been confirmed for other taxa (e.g., [110,111,112]).

The dissimilarity of hoverfly and bee populations increased with growing altitudinal distance. Similar results were detected for bees and wasps by Perillo et al. [91]. However, in our study, the change was more pronounced in hoverflies than bees: At the same distance along the gradient, the hoverflies community is more differentiated than that of bees. This can be explained by a difference in mobility and/or a greater specialization of hoverflies species with consequently narrower ranges of altitudinal distribution.

The strong differentiation of hoverfly and wild bee populations at different altitudes entails important conservation problems. The presence of well-differentiated populations of important pollinators at different altitudinal ranges highlights the need to plan conservation and monitoring interventions along the entire altitude gradient. Although at higher altitudes, at least in temperate areas, the anthropic impact is lesser, large-scale negative stressors should not be overlooked, in primis climate change. The list of species collected in the present research, together with their altitudinal distribution, are crucial to establish baselines for future research aimed at monitoring the spatial-temporal trend of these important pollinator taxa in order to understand the effects of climate change on biodiversity.

## Figures and Tables

**Figure 1 insects-13-00293-f001:**
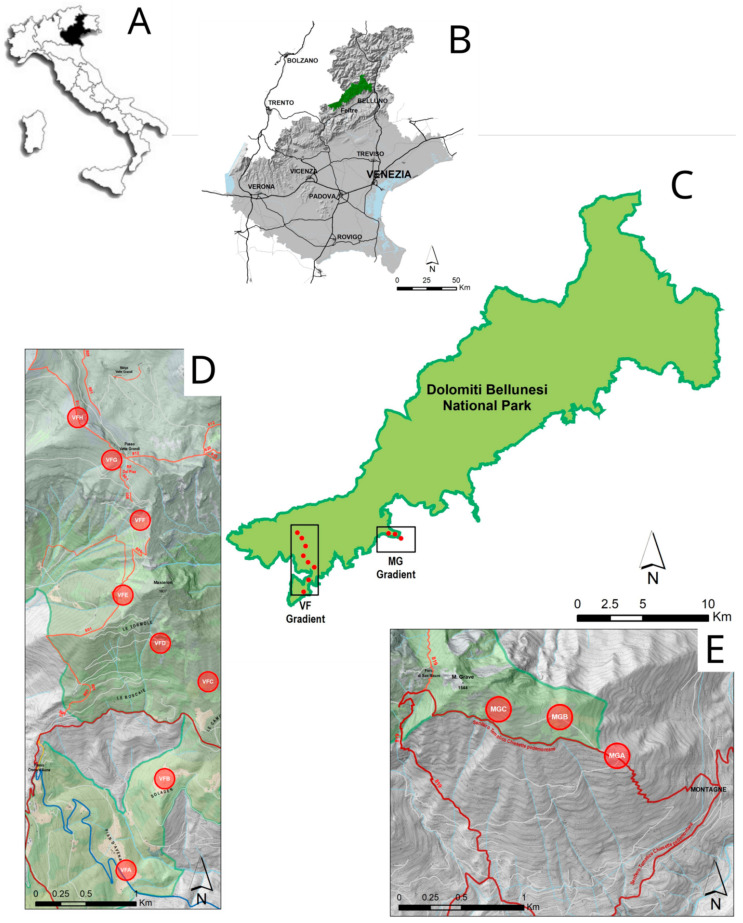
Maps of the research area: (**A**): Position of Veneto Region inside Italy; (**B**): Position of Dolomiti Bellunesi National Park in the Veneto Region; (**C**): Locations of the two altitudinal gradients in the Dolomiti Bellunesi National Park; (**D**,**E**) position of plot along the two altitudinal gradients (respectively, Vette Feltrine—VF—and Mount Grave—MG.

**Figure 2 insects-13-00293-f002:**
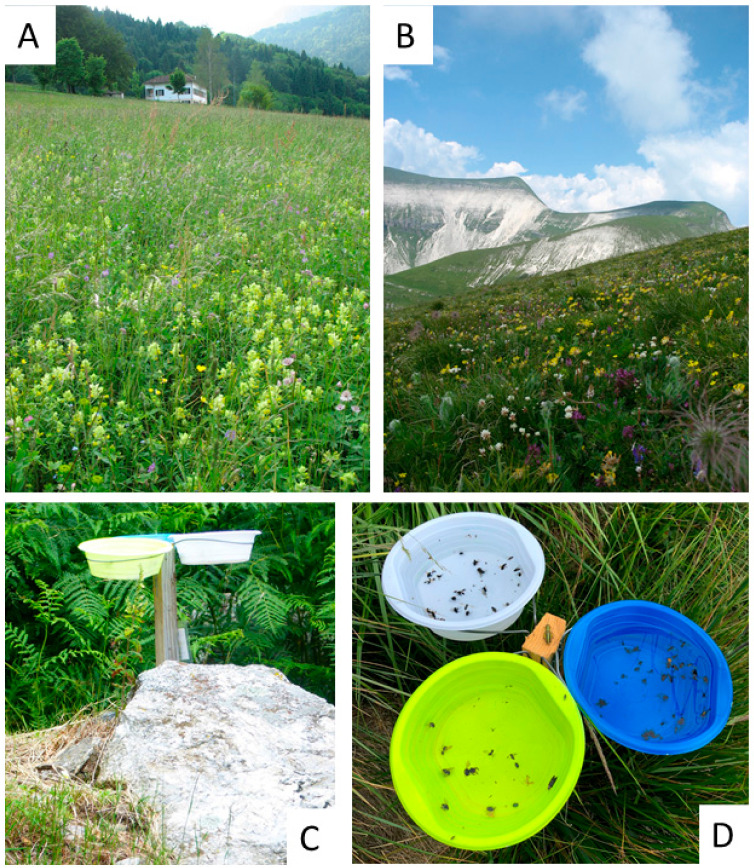
Two sampling points along the Vette Feltrine Gradient ((**A**): VFA, Pian d’Avena at 780 m; (**B**): VFH, Vette Grandi at 2130 m); the stick with the pan traps active (**C**,**D**).

**Figure 3 insects-13-00293-f003:**
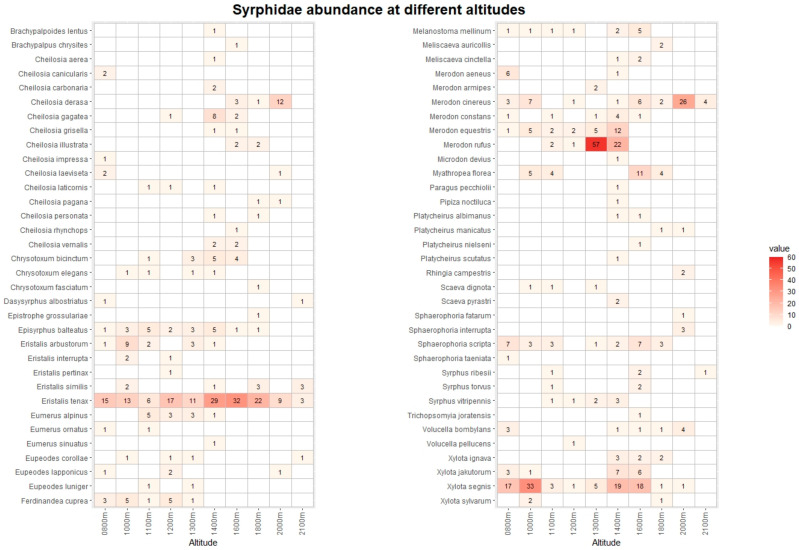
Heatmap with list of Syrphidae species and their abundance at different altitudes. The data from Vette Feltrine and Mount Grave are pooled. The intensity of color is proportional to specimen abundance.

**Figure 4 insects-13-00293-f004:**
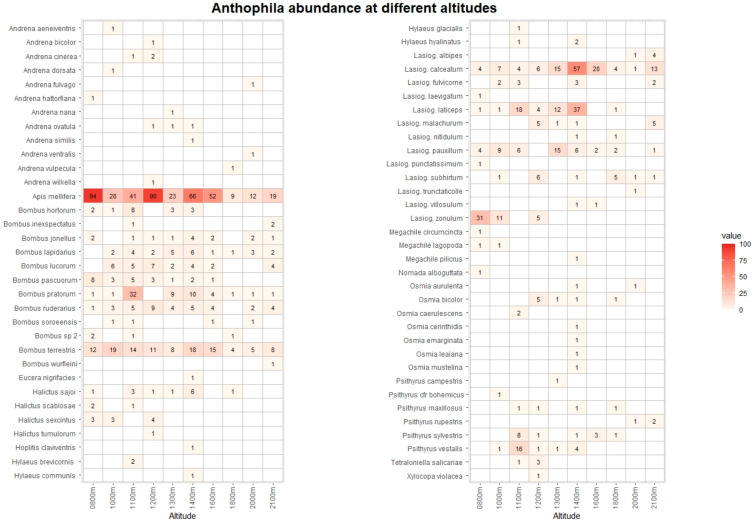
Heatmap with list of Anthophila species and their abunance at different altitudes. The data from Vette Feltrine and Mount Grave are pooled. The intensity of color is proportional to specimen abundance.

**Figure 5 insects-13-00293-f005:**
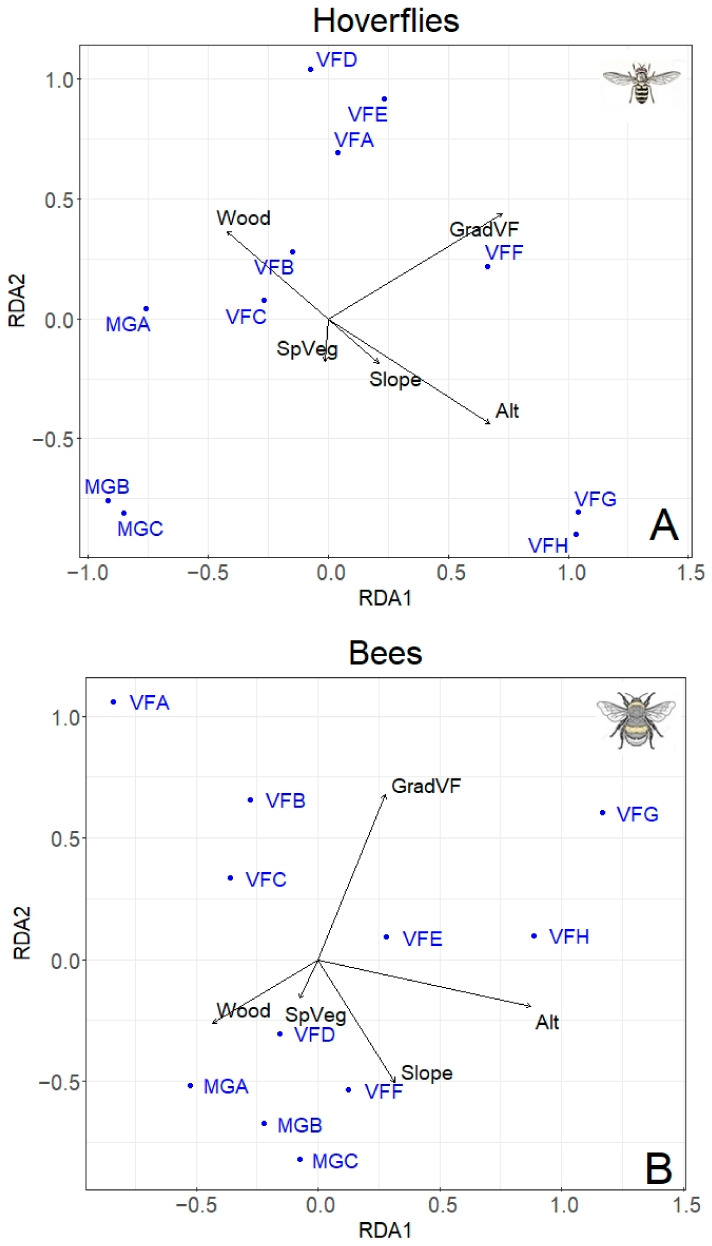
Redundancy Analysis (db-RDA) showing the effect of main predictors (altitude, type of gradient, slope, percentage of wood, total number of plant species) on hoverfly (**A**) and bee (**B**) populations at different altitudes. The sampling plots are in blues (for the abbreviations see Table 1). Proportion of variability explained by the two axes: Hoverfly (**A**) RDA1 = 27%; RDA2 = 19.38%; Bee (**B**) RDA1 = 28.46%; RDA2 = 18.81%.

**Figure 6 insects-13-00293-f006:**
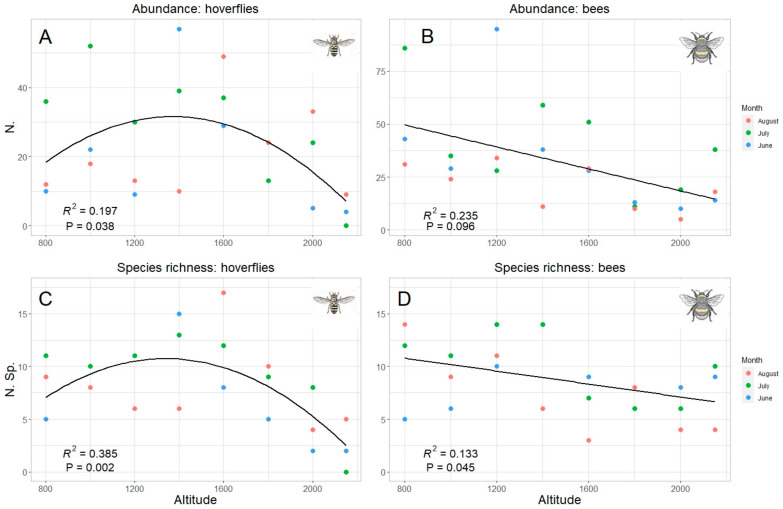
Relation between hoverfly (**A**,**C**) and bee (**B**,**D**) abundance (**A**,**B**) and species richness (**C**,**D**) and altitude; data for all months are pooled. The best model (linear or quadratic) has been selected in accordance with the results in Table 2.

**Figure 7 insects-13-00293-f007:**
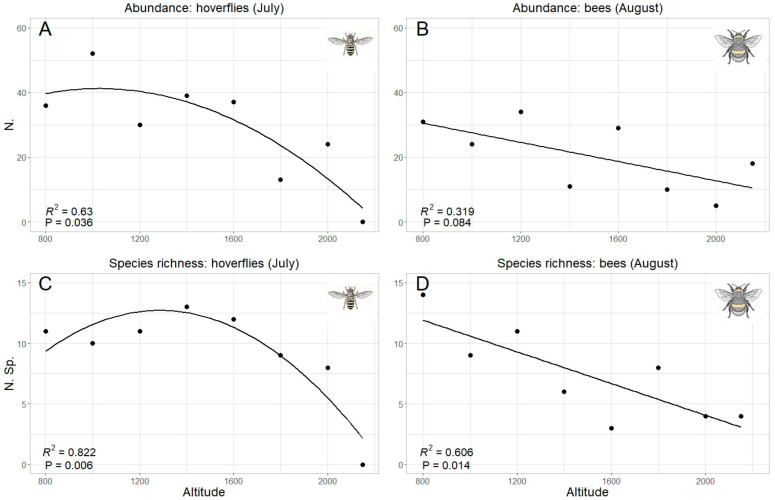
Relation between hoverfly (**A**,**C**) and bee (**B**,**D**) abundance (**A**,**B**) and species richness (**C**,**D**) and altitude; only the months with the best fit have been selected, in accordance with the results in Table 2.

**Figure 8 insects-13-00293-f008:**
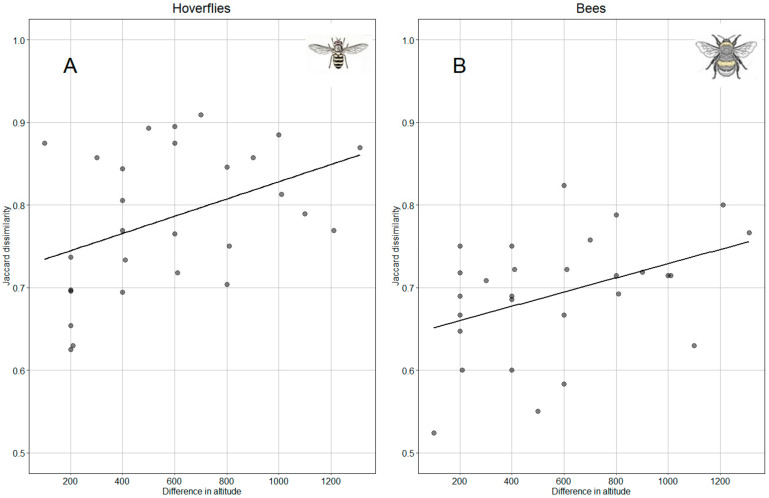
Relation of Jaccard dissimilarity index of hoverfly (**A**) and bee (**B**) with altitude.

**Table 1 insects-13-00293-t001:** Physical and environmental characteristics of the 11 sampling plots along the 2 altitudinal gradients: VF (Vette Feltrine) and MG (Mount Grave). For each sampling point, the geographical position, altitude, slope, and percentage of main habitat are reported.

Plot	Acronym	Gradient	Geographical Coordinates	Altitude (m)	Slope (°)	Type of Habitat (%)				
						Wood	Grassl.	Scree	Scrub	Other
Pian d’Avena	VFA	Vette F.	46.053101 11.841331	780	4	27	48	0	5	20
Soladen	VFB	Vette F.	46.060963 11.84648	956	5	3	90	0	7	0
Camogne	VFC	Vette F.	46.07009 11.853111	1210	26	1	72	0	27	0
Camogne Alte	VFD	Vette F.	46.073335 11.846008	1415	20	15	40	19	23	3
Cimetta	VFE	Vette F.	46.077279 11.842918	1635	17	38	42	0	17	3
Col Cesta	VFF	Vette F.	46.084766 11.844764	1790	35	13	40	15	26	6
Rifugio Dal Piaz	VFG	Vette F.	46.090258 11.841481	1990	15	0	97	2	0	1
Vette Grandi	VFH	Vette F.	46.09371 11.837018	2130	22	0	37	46	16	1
Casera al Pos	MGA	Grave M.	46.087296 11.939371	1075	7.5	54	27	0	16	3
Al Pian	MGB	Grave M.	46.090228 11.934328	1280	21	28	55	0	16	1
Monte Grave	MGC	Grave M.	46.091523 11.928041	1375	20	7	81	0	12	0

**Table 2 insects-13-00293-t002:** Results of best model (linear or quadratic) testing the effect of altitude on abundance and species richness of hoverflies and bees. The model has been tested both for all data pooled and for three sampling months separately. In bold, the model is statistically significant (*p* < 0.05). significance levels: 0.05–0.1; * 0.01–0.05; ** 0–0.01.

	Hoverflies		Bees	
	Abundance	Species Richness	Abundance	Species Richness
Total	**Quadratic** **R^2^ = 0.19 *p* = 0.042 ***	**Quadratic** **R^2^ = 0.385 *p* = 0.002 ****	**Linear** **R^2^ = 0.241 *p* = 0.009 ****	**Linear** **R^2^ = 0.137 *p* = 0.042 ***
June	Quadratic R^2^ = 0.238 *p* = 0.22	Quadratic R^2^ = 0.443 *p* = 0.1	Linear R^2^ = 0.238 *p* = 0.12	Quadratic R^2^ = 0.285 *p* = 0.19
July	**Quadratic** **R^2^ = 0.614 *p* = 0.04 ***	**Quadratic** **R^2^ = 0.767 *p* = 0.011 ***	Linear R^2^ = 0.234 *p* = 0.13	Linear R^2^ = 0.288 *p* = 0.098
August	Quadratic R^2^ = 0.07 *p*= 0.51	Quadratic R^2^ = 0.16 *p* = 0.62	Linear R^2^ = 0.335 *p* = 0.078	**Linear** **R^2^ = 0.606 *p* = 0.014 ***

## Data Availability

Data are contained within the article and Appendix A.

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
