# Peer review of "Different Distribution Patterns of Hoverflies (Diptera: Syrphidae) and Bees (Hymenoptera: Anthophila) Along Altitudinal Gradients in Dolomiti Bellunesi National Park (Italy)"

_insects, 2022, doi:10.3390/insects13030293_

Round 1
Reviewer 1 Report
This paper presents results of a local monitoring study on species diversity and composition of bees and hoverflies in the Italian Alps. The reported patterns are interesting, though not truly surprising. However, the paper is far too long, in relation to its modest information content. In particular, the references list is excessive and contains, as the main text, multiple entries that stray far apart from what the paper is all about. The paper is about patterns at one point in time, i.e. the status quo.
Since there are no data (yet) on temporal changes (e.g. due to climate change) reference to this (important) topic should be much reduced and underpinned only with a handful of RECENT up-to-date references. The manuscript lists many papers 10-20 years old on climate change effects such as upslope movement of species. This can and should be massively condensed, especially since this is a vivid field of research and many of these older papers are outdated by now (in the sense that newer studies are available based on more/better data).
The paper lumps together data from two sub-regions, but in one of these no real elevational gradient was studied, rather just 3 sites were available. Take them out!! In the analysis of elevational patterns these are meaningless anyway. And in the analysis of species composition by ordination, they just showed to be “different”, as one would expect. I strongly suggest to focus only on the data from the VF gradient, which is a real elevational gradient (though incomplete, lacking low-elevation sites).
I am also concerned with many aspects of the statistical data analysis. For species composition, I suggest to include abundance information, e.g. do a dbRDA or a CAP, based on Bray-Curtis matrices. Species abundances should be sqrt or Hellinger/Chord transformed (Legendre, P., & Gallagher, E. D. (2001). Ecologically meaningful transformations for ordination of species data. Oecologia, 129(2), 271-280.) You may consider during revision whether (or not?) for the analysis of compositional diversity it could be worthwhile to treat monthly replicates as separate data points.
I strongly argue against using DCA as a method, since this algorithm arbitrarily sections gradients into “segments”, as a kind of artefact. This is simply not necessary. Use NMDS (for uconstrained ordination( or dbRDA / CAP (for constrained ordination) instead.
For the regression of richness or abundance vs. elevation, I see no reasoning for segregating these analyses according to sampling months. Samples from the same site, but taken in different months violate the basic assumption of stochastic independence which underlies any OLS regression. Hence, the regression results you have reported are invalid – either you have to account for site ID as a random effect, or (preferably, given the small data body!) you aggregate samples per site across monthly replicates. Or you might check for a month x elevation interaction, IF you should have a well-founded testable hypothesis in that regard.
You have really few sites available for analysis (i.e. few degrees of freedom), but conducted many statistical tests. This has a high risk of inflated Type I error rates (due to the Bonferroni inequality). All tests need to be adjusted accordingly, e.g. using a False Discovery Rate approach (Verhoeven, K. J., Simonsen, K. L., & McIntyre, L. M. (2005). Implementing false discovery rate control: increasing your power. Oikos, 108(3), 643-647.) Please clearly report which results ‘survive’ after such a control, and which might ‘disappear’.
Leave out your attempts to identify patterns among subgroups of both focal taxa (L 226-237). Given your small data body, nobody could even expect that any meaningful patterns would emerge, and this was obviously also not the case. This does not mean that certain guilds or functional groups might differ in their elevational responses. This simply indicates the power of your small sub samples was far too small to arrive at any serious results.
I also noted that area size of the elevational habitats (‘belts’) was completely neglected in the paper. Mid domain effects (MDE) have largely been ignored, but the hump-shaped elevational pattern observed in hoverflies could simply be due to an MDE, i.e. a result of geometric constraints (with no further ‘ecological’ drivers behind). This needs to be tested, i.e. use appropriate models that compare your results with what one might expect from geometry alone. For example: https://rdrr.io/cran/rangemodelR/man/rangemodelR-package.html I do NOT state that the pattern is an MDE effect alone, but this needs to be demonstrated adequately.
A conceptually most important paper the authors should read and incorporate:
Rahbek, C., Borregaard, M. K., Colwell, R. K., Dalsgaard, B. O., Holt, B. G., Morueta-Holme, N., ... & Fjeldså, J. (2019). Humboldt’s enigma: What causes global patterns of mountain biodiversity? Science, 365(6458), 1108-1113.
Finally, the paper is in need of VERY thorough linguistic polishing in regard to grammar and style.
Author Response
We would like to thank the reviewer for taking the time to read our manuscript and for the constructive comments that have been advanced. We improved the manuscript following his suggestion: A detail of the changes made can be found in the attached file

Reviewer 2 Report
This is a very nice work on community structure for two groups of interest bees and hoverflies. I really enjoyed reviewing this paper. This paper brings more understanding about the role of altitude on bee and hoverflies assemblages as key drivers. I highly recommend that this paper could be published in Insects. All the data analysis are appropriate, the results and discussion part are well written.
For me, one of the main issues of the paper is the timeline inconsistencies of the protocol. I suppose that the authors did a very fine job and they are fully involved in their project. But, for the reader (as first read), it is very difficult to understand when the data are collected, especially them for the insect distribution. Why take data of temperature in 2019 against 2013 insect collection, or maybe I had not understood in the paper? I suggest here a restructuration of the experience timeline as readers could better understand and apprehend the timeline experience.
I also encourage authors to write a paragraph in the discussion part why they did not choose to use net beside pan traps technique as these methods are considered now as gold standard to perform bees and hoverflies collections for that kinds of research questions.
Specific comments :
43 : can we talk about phenomenon here ? Another word should be more appropriate.
49-51 : In which year the annual value of pollination was assessed ?
52-54 : Pesticides are clearly one of the main drivers of the pollination crisis, but I think that the big one is the alteration of landscapes which reduces the availability of floral and nesting resources (see Potts et al., 2010) and diminishing tolerance threshold against the pesticide or disease exposition (Di Pasquale et al., 2013). This could be reinforced in this paragraph. I completely agree with the following statement about the impact of the climate change.
72 : I do not understand the meaning of “‘800”, is it altitude or years ?
73 : One bracket is missing
87-88 : reformulate the sentence
95 : When the biomonitoring ends ?
100 : Anthophila (Hymenoptera) and Syrphidae (Diptera)
103 : same as line 100
114 : Could you add some GPS coordinates for the study design ?
119 : What are the altitudes differences for the second gradient ?
112 to 139 : I would suggest that the study design lines are structured by whole sentence and not by numerous point. Also in the other part of the manuscript.
129: By which tool or institution do you obtain the map file ? This must be added at this point.
Figure 1 legend : Scale and orientation are missing.
150 : I suggest “hoverflies (Diptera:Syrphidae) and bees (Hymenoptera:Anthophila)”
151 to 155 : this part must be merged in the introduction part.
156 : “were sampled by colored pantraps (white, blue and yellow)”
163 : Need a comma after “plot” and “in each sampling point”
165 : I think that the proximity of each three colored bowls on each sampling point could influence their attraction as reported by Droege 2010 (Droege S, Tepedino VJ, Lebuhn G, Link W, Minckley RL, Chen Q, Conrad C (2010) Spatial patterns of bee captures in North American bowl trapping surveys. Insect Conservation and Diversity, 3, 15–23.) , it could be mentioned in the Discussion part. Why do you choose 60 cm as height ?
170 : It is difficult to understand which year of sampling ?
Figure 2 : I suggest better quality pictures if possible, or maybe it is a problem from the passage to .pdf format.
181 : Can you give us the accession number of the barcoded specimens ? Can you prepare a supplementary information on the used methods?
228 : microphabitat è microhabitat
244 : Why a “6” ? & 254 why a “7 ?”
265 : references needed for ggplot2 and ggpubr, I think that all R packages should be italicized
Table 2 & 3 : Same issue than Figure 2
410 : was
Author Response

(The authors gave the same response as above.)

Reviewer 3 Report
Present paper reports altitudinal distribution of hoverflies (Diptera:Syrphidae) and bees (Hymenptera:Anthophila) in Dolomiti Bellunesi National Park in Italy. Authors have performed pan-trap collections during summer months at different altitudes at two different gradients. The data in itself is valuable and of wide interest to many scientists. However, there are number of concerns regarding this manuscript. Material and method section is missing many details (please see in the line by line comments). The results section also can be improved, at the moment it is rather hard to follow it. The quality of tables, figures should be improved. It is impossible to read some of them. The discussion part should be rewritten as in present form it mainly present a literature overview that should be moved to introduction part. The authors should provide more feedback about their own results and their meaning. It is also important to discuss factors that potentially could've influenced their results.
Please see specific line by line comments attached.

Author Response

(The authors gave the same response as above.)

Reviewer 4 Report
Line 31: dominates
Line 34: represents
Line 54: in the European Union
Line 63: According to
Line 243: two approaches were used
Line 305: a higher association with vector
Line 330: In bold the model is statistically significant.
Author Response

(The authors gave the same response as above.)

Round 2
Reviewer 1 Report
The authors have addressed most (not all) points raised during review. They discarded some suggestions. I disagree with that position, but this is a matter of editorial policy. For example, I maintain that with the application of (very!) many statistical tests on the same corpus of data, these tests cannot be regarded as stochastically independent and thus a correction of significance levels according to the Bonferroni inequality is appropriate and needed.
But if the editors take a different view: fine.
Overall the paper has been substantially improved. Since the authors maintain to present part of their data split between different time horizons, this aspect of possible phenological vartiation in diversity patterns should be included explicitly as a third research question at the end of the introduction.
I came across a few minor points.
L 74: I am unaware of any study to show an increase of with elevation in Lepidoptera in the European Alps! At high elevations diversity and abundance usually decrease, as shown for example in the study by Beck et al which you cite. Also, for butterflies: Leingärtner, A et al. (2014). Species richness and trait composition of butterfly assemblages change along an altitudinal gradient. Oecologia, 175(2), 613-623.
Pellissier, L., et al. (2012). Shifts in species richness, herbivore specialization, and plant resistance along elevation gradients. Ecology and Evolution, 2(8), 1818-1825.
Pellissier, L., et al. (2013). Phylogenetic alpha and beta diversities of butterfly communities correlate with climate in the western Swiss Alps. Ecography, 36(5), 541-550.
L 164: probably should better read ‘steep slope’
L 175: probably you rather mean ‘sampling dates’?
L 381: probably should read ‘efficiency’
L 387-389: sentence very difficult to understand. Please re-phrase.
L 391 ff: for sure these surveys were 'incomplete'. If that is an issue for your study, then perform a proper coverage analysis using the iNext package in R. If this is not an issue of relevance: leave out all these discussions about "completeness" since collections of mobile animals, especially with such a low level of spatial and temporal replication, are NEVER "complete". As long as the sampling intensity and bias is the same across the samples to be compared, this entire topic is a non-issue. If, on the other hand, you feel this coverage issue to be relevant, then (a) pose this as a research question in the introduction and (b) perform a rigorous (statistical) analysis (e.g. using iNext).
L 402 should read ‘research’
L 409: moths are NOT a monophyletic taxon.
